# NOV/CCN3 Promotes Cell Migration and Invasion in Intrahepatic Cholangiocarcinoma via miR-92a-3p

**DOI:** 10.3390/genes12111659

**Published:** 2021-10-21

**Authors:** Tingming Liang, Lulu Shen, Yaya Ji, Lin Jia, Yuyang Dou, Li Guo

**Affiliations:** 1Jiangsu Key Laboratory for Molecular and Medical Biotechnology, School of Life Science, Nanjing Normal University, Nanjing 210023, China; tmliang@njnu.edu.cn (T.L.); 191202064@stu.njnu.edu.cn (L.S.); 171202083@stu.njnu.edu.cn (Y.J.); 201202100@njnu.edu.cn (L.J.); 2Department of Bioinformatics, Smart Health Big Data Analysis and Location Services Engineering Lab of Jiangsu Province, School of Geographic and Biologic Information, Nanjing University of Posts and Telecommunications, Nanjing 210023, China; 1020173016@njupt.edu.cn

**Keywords:** NOV/CCN3, miR-92a-3p, cell migration, invasion, intrahepatic cholangiocarcinoma

## Abstract

Intrahepatic cholangiocarcinoma (ICC) is a common type of human cancer with a poor prognosis, and investigating the potential molecular mechanisms that can contribute to gene diagnosis and therapy. Herein, based on the recently concerned vertebrate-specific Cyr61/CTGF/NOV (CCN) gene family because of its important roles in diverse diseases, we obtained NOV/CCN3 to query for its potential roles in tumorigenesis via bioinformatics analysis. Experimental validations confirmed that both NOV mRNA and protein are up-regulated in two ICC cell lines, suggesting that it may promote cell migration and invasion by promoting EMT. To elucidate the detailed regulatory mechanism, miR-92a-3p is screened and identified as a negative regulatory small RNA targeting NOV, and further experimental validation demonstrates that miR-92a-3p contributes to NOV-mediated migration and invasion of ICC via the Notch signaling pathway. Our study reveals that NOV may be a potential target for diagnosing and treating ICC, which will provide experimental data and molecular theoretical foundation for cancer treatment, particularly for future precision medicine.

## 1. Introduction

Intrahepatic cholangiocarcinoma (ICC), a refractory liver malignancy, is one of the most common cancers, and patients diagnosed with ICC are challenging to cure with poor prognosis [1]. The median survival time of patients is less than 24 months [2]. Only 15% of patients are identified as a resectable disease [3] because it is difficult to obtain an early diagnosis, which mainly derives from asymptomatic or non-specific clinical symptoms [4]. Currently, surgical resection is still the only method to cure this disease, but only 30–60% of patients are candidates due to locally advanced or metastatic disease or frailty [5]. Following surgical resection, the median survival time is 25 months, and the 5-year survival is 20–40% [5]. It is critical to screen potential molecular markers for predicting therapeutic responses that would further contribute to studying molecular mechanisms underlying ICC, particularly for its effective management. 

Many studies have revealed that some abnormally expressed genes have crucial roles in the occurrence and development of ICC. For instance, *eIF3C* and *KI67* may be valuable predictors of survival and recurrence of ICC patients [6], NR4A2/OPN/Wnt signaling axis may be a pivotal executor of hepatic stellate cells instigated cancer-promoting roles in ICC [7], typing *FGFR2* translocation can determine the response to targeted therapy of intrahepatic cholangiocarcinoma [8], and DKK-1 combined with CA 19-9 may be the potential diagnostic and prognostic marker that may be better than CA 19-9 alone [9]. These studies strongly indicate that some key genes may be potential markers with diagnostic and prognostic values, implying their roles in cancer pathophysiology. Similarly, some genes have been validated as having potential roles in ICC development. Blockade of *CXCL12/CXCR4* signaling can inhibit ICC progression and metastasis through inactivation of canonical Wnt pathway [10], lncRNA MT1JP has a protective role via regulating miR-18a-5p/FBP1 axis [11], and crenigacestat, a selective NOTCH1 inhibitor, can reduce ICC progression via blocking VEGFA/DLL4/MMP13 axis [12]. The study that focuses on crucial genes in cancer will provide more important references by elucidating the detailed molecular mechanisms. 

Numerous studies have found that the CCN gene family may also have potential roles in various diseases. Six homologous members have been identified, mainly including *CCN1 (Cyr61)*, *CCN2 (CTGF)*, *CCN3 (NOV)*, *CCN4 (WISP1)*, *CCN5 (WISP2)*, and *CCN6 (WISP3)*, which is only detected in vertebrates as a vertebrate-specific gene family [13]. They are implicated in the pathological processes of fibrosis [14], development [13], and inflammation [15]. This family has been validated with important roles in cancer progression [16,17,18,19]. CCN2 is a prognostic marker [20], and CDK/CCN and CDKI alterations may contribute to cancer prognosis and therapeutic predictivity [21]. CCN1 is identified as a tumor suppressor in non-small cell lung cancer [22], and it is also can be used to predict survival with endometrial cancer of endometrioid subtype [23]. Vitamin D-mediated regulation of CCN genes may be an adjuvant therapy for cancer and fibrosis [24], and CDK/CCN and CDKI alterations for cancer prognosis and therapeutic predictivity [21]. Although relevant studies have demonstrated the potential crucial roles of the CCN gene family in cancer, fewer studies have examined the detailed biological roles in ICC occurrence and development. Given the poor prognosis associated with ICC, it is quite necessary to understand and validate the potential biological roles of the CCN family to provide relevant references for precision treatment.

Herein, based on homologous members in the CCN gene family, we screened and verified abnormally expressed NOV (Nephrblastoma overexpressed, also known as CCN3) via pan-cancer analysis to perform further experimental validation. We found that NOV overexpression may be derived from miR-92a-3p regulation, contributing to NOV-mediated migration and metastasis of intrahepatic cholangiocarcinoma cells. Our study reveals that NOV is a potential target for pre-ICC diagnosis and treatment, providing experimental evidence and a molecular theoretical foundation for cancer treatment, particularly for future precision medicine.

## 2. Materials and Methods

### 2.1. Data Resources

We first analyzed expression patterns of the six homologous gene members in the CCN gene family for their expression patterns via a pan-cancer analysis using high-throughput sequencing data in the Cancer Genome Atlas (TCGA) (https://tcga-data.nci.nih.gov/tcga/, accessed on 28 January 2021) with the “TCGAbiolinks” package [25] (http://doi.org/10.1093/nar/gkv1507, accessed on 28 January 2021). To screen candidate crucial genes associated with ICC, an integrative analysis was performed using Starbase [26] and GEPIA [27] databases. Differentially expressed genes were mainly analyzed using DESeq2 [28]. The genes were believed as abnormally expressed if |log_2_FC| > 1.5 and padj < 0.05, or if |log_2_FC| > 1.2 and *p* < 0.05 based on further paired analysis. 

### 2.2. Cell Culture and Grouping

To further understand the biological function of screened genes, cell lines (HIBEC, human ICC HCCC-9810 and RBE), were purchased from Shanghai Institute of Cell Biology, Chinese Academy of Sciences (CAS; Shanghai, China). All cells were grown in RPMI-1640 medium containing 10% fetal bovine serum (FBS) (both obtained from Gibco, Grand Island, NY, USA), 100 µg/mL streptomycin and 100 U/mL penicillin (HyClone, Logan, UT, USA). The cells were stored in a 5% CO_2_ atmosphere at 37 °C, and then were digested with 0.25% pancreatin-ethylene diamine tetraacetic acid and diluted in a ratio of 1:3 after reaching 80% confluency.

### 2.3. Construction of Plasmid and Transfection

The full-length of NOV cDNA (Ensembl: ENSG00000136999) was cloned into pcDNA3.0 expression vector and then transfected into HCCC-9810 or RBE cells. The cells were seeded in six-well plates. When the fusion rate reached about 85%, the cells were then transfected using NOV empty vector and overexpression vector respectively in line with Lipofectamine™ 3000 instructions (Thermo Fisher Scientific, Waltham, MA, USA). After 48 h, the residual liquid in each well was replaced with DMEM 1640 complete medium. The cells were returned to the incubator at 37 °C, 5% CO_2_, and 95% humidity. After 48 h, the transfected cells were used to determine the expression level of NOV using qRT-PCR and Western blot assays, respectively. Empty vector-transfected cells were used as controls.

### 2.4. RNAi and Transfection

Three designed NOV siRNAs and negative control siRNA were synthesized by Biotend, China. The sequences of siRNAs were as follows: 5′ GCACCAAGAAGUCACUCAAdTdT 3′ (sense) and 5′ UUGAGUGACUUCUUGGUGCdTdT 3′ (antisense) for si-NOV-1, 5′ TAACTGCCCAGCTCCAAGAAdTdT 3′ (sense) and 5′ GAACCCCATACCACAGCTCTdTdT 3′ (antisense) for si-NOV-2, 5′ GAGAUAACUGUGUGUUCGdTdT 3′ (sense) and 5′ UCGAACACACAGUUAUCUCdTdT 3′ (antisense) for si-NOV-3. The cells were cultured with six-well plates and then transfected with siRNAs using Lipofectamine 3000 (Thermo Fisher Scientific, Waltham, MA, USA) according to the manufacturer’s protocol. After 48 h, the cells were harvested for the subsequent experiments.

### 2.5. Colony Formation Assay

The ICC cells (HCCC 9810 and RBE) were suspended in RPMI-1640 medium with a final concentration of 500 cells per microliter and then seeded into six-well plates (Corning Inc., Corning, NY, USA). The culture medium (containing 10% FBS) was replaced every three days. When a signal colony contained >50 cells, The cells were fixed and stained. Following that, the colony-forming units were stained with crystal violet and then photographed by microscope (Nikon Eclipse Ts2R; Nikon, Tokyo, Japan). The results were repeated in three independent experiments.

### 2.6. Cell Viability Assay

Cell viability test was performed in triplicate using a commercially available kit (Cell Counting Kit-8, CCK-8) (Tokyo, Japan). The cells were diluted to single-cell suspension and then seeded into 96-well plates (1 × 10^3^ cells/well) with 100 µL culture medium. After 12 h incubation, the cells were treated with siRNAs (100 nm) or NOV overexpression vector (1000 ng) for 48 h. At the time point (0, 24, 48, 72, and 96 h), CCK-8 solution (10 µL) was added to each well and incubated for 3 h. The cell viability was calculated as absorbance of solutions that were detected at 450 nm with a microplate reader (Epoch, Biotech, Winooski, VT, USA). The results were represented as an average of five parallel samples. All related experiments were repeated at least three times.

### 2.7. In Vitro Migration and Invasion Assays

For wound-healing assay in vitro, cells (about 1 × 10^6^ cells) were seeded into six-well plates, and cells were incubated at 37 °C until reaching at least 90% confluence. Wounds were created via scratching cell monolayers using a 200 μL plastic pipette tip, and then they were incubated in a fresh medium that contained 1% fetal calf serum. Cell migration into wound was monitored in a serum-free medium, which was photographed by a fluorescence microscope at 0 and 48 h, respectively. Transwell filters (8 μm, Greiner Bio-One, Frickenhausen, Germany) were used to detect cell migration and invasion. HCCC-9810 (1 × 10^5^) and RBE cells (1 × 10^5^) in 200 μL serum-free mediums were, respectively, added to the upper chamber containing an uncoated or Matrigel (Corning Life Sciences, Manassas, VA, USA)-coated membrane. The lower chamber was filled with 800 μL basal medium containing 10% FBS. After 48 h of incubation at 37 °C in a humidified 5% CO_2_ incubator, the cells migrated to the lower compartment were fixed with methanol and then stained using crystal violet. Migrated or invaded cells were counted in three randomly chosen fields in each well. Imaging and cell counting were performed at 10× magnification with a fluorescence microscope. The relevant experiments were performed in triplicate.

### 2.8. Quantitative Real-time RT-PCR

Total RNAs of HIBEC, HCCC-9810, and RBE cells were extracted using TRIzol reagent (Invitrogen, Carlsbad, CA, USA). cDNA was then generated from the total RNAs using a reverse transcription kit (Takara, Dalian, China). Gene expression level was measured using an SYBR green kit (Yeasen, Shanghai, China) by q-PCR (Lightcycler96, Roche, Basel, Switzerland). Primers sequences of NOV (sense: 5′ TAACTGCCCAGCTCCAAGAA 3′, antisense: 5′ GAACCCCATACCACAGCTCT3′). GAPDH (sense: 5′ CGGAGTCAACGGATTTGGTCGTATTGG 3′, antisense: 5′ GCTCCTGGAAGATGGTGATGGGATTTCC 3′) and U6 snRNAs (5′ CTCGCTTCGGCAGCACATATACT 3′, antisense: 5′ ACGCTTCACGAATTTGCGTGTC 3′) were used to normalize mRNA and miRNA levels, respectively. The expression of related genes was analyzed by the 2^−ΔΔCt^ method (the control group was normalized).

### 2.9. Western Blot Analysis

HIBEC, HCCC-9810, and RBE cells were transfected with siRNAs (100 nm) or NOV overexpression vector (1000 ng) for 48 h. Then the cells were harvested and lysed in RIPA buffer (Beyotime Institute of Biotechnology, Beijing, China) supplemented with protease inhibitor (Roche Applied Science, Indianapolis, IN, USA) at 4 °C for 15 min. The cells were centrifuged with a microcentrifuge at 12,000× *g* for 15 min at 4 °C to collect the supernatant. The protein concentration was then measured using bicinchoninic acid (BCA) assay kit (Beyotime). After boiling for 5 min in 1× SDS sample buffer, a total of 30 µg of protein/lane was subjected to SDS-polyacrylamide gel electrophoresis (PAGE) and then electrophoretically transferred to polyvinylidene difluoride membranes (Millipore, Bedford, MA, USA). After blocking using 5% skim milk for 1–2 h, the membranes were incubated with primary antibodies against Snail (1:1000), MMP9 (1:1000), vimentin (1:2000), N-cadherin (1:10000), E-cadherin (1:4000), NOV (1:5000) and GAPDH (1:10000) at 4 °C overnight. After washing with Tris-buffered saline and Tween-20 (TBST) buffer three times (5 min per time), the blots were re-probed using secondary antibodies conjugated to horseradish peroxidase for 1 he. The protein bands were then observed using enhanced chemiluminescence and visualized with a Gel Doc 2000 (Bio-Rad, Hercules, CA, USA). Sch B and Rhodamine 123 (Rho 123) were obtained from Sigma-Aldrich (St. Louis, MO, USA). Sch B was dissolved in dimethyl sulfoxideto create a 100 mm stock solution. The control groups were treated with equal volumes of DMSO. Primary antibodies against Snail, MMP9, vimentin, N-cadherin, E-cadherin, GAPDH, and secondary antibodies (goat anti-rabbit) were all purchased from Proteintech (Wuhan, China). NOV antibody was purchased from Abcam (Cambridge, United Kingdom).

### 2.10. Luciferase Reporter Construction

A 3′-UTR region (1302 bp) of human NOV gene, containing a putative target site for miR-92a-3p, was identified from PITA (https://genie.weizmann.ac.il/pubs/mir07/mir07_prediction.html, accessed on 18 March 2021), miRmap [29], miRanda [30] and TargetScan database [31] and amplified by PCR (miR-92a-3p primers: sense: 5′ GCGTATTGCACTTGTCCCG 3′; antisense: 5′ AGTGCAGGGTCCGAGGTATT 3′). The target fragment was used to insert between XhoI and NotI, in the pmiR-RB-Report^TM^ vector. We also mutated 7 bp of the miR-92-3p binding site from GTGCAAT to CACGTTA. Wild-type (WT) and mutant (Mut) insert sequences were confirmed and then transfected into HCCC-9810 cells. The cells were lysed, and supernatants were collected for further analysis. Relative luciferase activities were detected with commercial kits (E1910, Promega, Madison, WI, USA) and examined on a Luminoscan Ascent (Thermo Fisher Scientific, San Jose, CA, USA).

### 2.11. Statistical Analysis

Numeric data were presented as mean ± standard deviation (SD) for multiple samples, and some methods were used to perform hypothesis testing in relevant analyses, mainly including Wilcoxon rank-sum test, paired *t*-test, a Student’s *t*-test or one-way ANOVA analysis followed by Fisher’s LSD post-hoc test, etc. All experiments were repeated at least three times. A *p*-value < 0.05 was considered statistically significant. All analyses were performed using the R programming language (version 3.4.3).

## 3. Results

### 3.1. Expression Patterns of CCN Family and Screening Crucial Genes in ICC

To ascertain the potential expression patterns of the six members in the CCN gene family, expression analysis was performed in a variety of cancer types. A total of 18 cancer types were analyzed, and five genes were identified except for Cyr61. Most of them were abnormally expressed in different tissues (Figure 1A), especially for WISP2, which was down-regulated in 10 different tissues but up-regulated in KIRC. WISP1 was significantly up-regulated in eight tissues but down-regulated in UCEC. These dynamic expression patterns indicated that these homologous genes might have played important critical roles in cancer pathology, with the exception of NOV and WISP3, which had abnormal expression patterns in cholangiocarcinoma (also named CHOL in TCGA).

Despite the homology of these genes, which suggests a close association in biological processes, they demonstrated dynamic expression distributions in tumor and normal samples in ICC (Figure 1B). CTGF was the dominantly expressed gene, followed by NOV, whereas WISP3 and WISP2 had lower expression levels. To further understand the detailed expression patterns revealed by paired analysis, the two abnormally expressed genes were performed expression analysis across diverse cancer types (Figure 1C) using TCGA CHOL RNA sequencing data. Paired analysis revealed that NOV expression patterns were more deregulated in several cancers, while WISP3 expression patterns were more divergent than in total samples. Both analyses revealed that NOV and WISP3 were abnormally expressed in cholangiocarcinoma, but WISP3 was identified to be down-regulated and expressed at a lower level. Therefore, we finally screened NOV as a potential crucial gene associated with ICC to perform further experimental validation to ascertain its roles in biological processes, especially in the occurrence and development of cancer. 

### 3.2. Experimental Validation Shows High Expression of NOV

Although the paired analysis indicated that the NOV gene could be deregulated in a variety of cancer types, significant expression patterns were detected only in BRCA (down-regulated) and CHOL (up-regulated) cancers (Figure 2A). The expression patterns implied its spatiotemporal expression across diverse tissues. 

To further understand the detailed expression patterns in cells, we examined the expression levels of HCCC-9810 and RBE cells. Compared with BEC cells (normal human biliary epithelial cells), NOV mRNA in both HCCC-9810 and RBE cells were significantly up-regulated 15-fold (*p* < 0.001) and 23-fold (*p* < 0.001), respectively (Figure 2B). Simultaneously, NOV protein also revealed similar results (Figure 2C), indicating that NOV was also significantly overexpressed in cell lines and bioinformatics analysis. Numerous relevant studies have demonstrated that the potential roles of NOV in cancer. For instance, NOV may be an attractive target for therapeutic intervention in the alveolar subtype [32], and NOV may have a tumor-suppressive role in prostate cancer [33]. It is critical to elucidate the potential role of NOV in tumorigenesis.

### 3.3. NOV Is Involved in Cell Proliferation Process

Transfection efficiency of NOV overexpression plasmid was validated in HCCC-9810 and RBE cells, respectively. The results indicated that relative expression levels of NOV mRNA were increased (*p* < 0.001), and NOV protein also demonstrated similar results (*p* ≤ 0.01) (Figure 3A,B). Concurrently, we also designed siRNA targeting NOV and NOV silencing to study the potential effect of NOV on cell proliferation. We finally selected si-NOV-1 and si-NOV-2 to perform additional functional validation (Figure 3C,D).

The multiplication rate of cell groups with NOV overexpression was greater than that in the negative control group at 96 h using CCK-8 assay, but no significant difference was detected than that in the control group (Figure 4A). Compared with the negative control group, multiplication rates of cell groups of si-NOV-1 and si-NOV-2 were reduced at 48 h, but no significant difference was detected (Figure 4B). Similarly, colony formation assay showed that the number of colonies with overexpression and silence of NOV was not significantly increased or inhibited than that in the control group (Figure 4C,D). These findings indicated that NOV had no significant effect on the cell proliferation process in HCCC-9810 and RBE cells.

### 3.4. NOV Promotes the Cell Migration and Invasion

To understand the potential effect of NOV on cell migration ability, wound healing and transwell chamber assays were used to detect NOV overexpression and silencing. NOV overexpression may accelerate wound healing, indicating that NOV expression could increase cell migration ability (Figure 5A). On the other hand, NOV silencing could enlarge the wounding region in si-NOV-1 and si-NOV-2 groups, indicating a decreased ability of cells to migrate (Figure 5B). This was mainly derived from NOV silencing that might inhibit cell migration. These results implied that NOV promoted cell migration, which may be crucial in the occurrence and development of cancer. 

Simultaneously, NOV overexpression could promote cell migration as measured by transwell chamber assay (Figure 5C), and the number of passing through polycarbonate membrane cells was significantly higher than that in the control group at 48 h. However, compared with the si-NC group, migration rate and the number of passing through polycarbonate membrane cells were significantly reduced si-NOV-1 and si-NOV-2 treatment (Figure 5D). These findings revealed the potential role of NOV in cell migration, and NOV can promote the migration of cholangiocarcinoma cells. 

To further verify the invasive role of NOV, overexpressed plasmids were transfected into HCCC-9810 and RBE cells, and a transwell invasion assay was performed after 48 h. Similar to migration results, overexpression of NOV could promote ICC cell invasion abilities (Figure 6A), whereas NOV silencing could impair their the invasion abilities. These findings proved confirmed that NOV could promote invasion ability of cells, indicating the potential role of NOV in cancer pathology (Figure 6B). Furthermore, Western blot analysis revealed showed that NOV overexpression could down-regulate E-cadherin, an epithelial marker protein, and simultaneously up-regulate vimentin, N-cadherin, Snail and MMP9 (Figure 6C). However, NOV silencing could up-regulate E-cadherin and down-regulate vimentin, N-cadherin, Snail and MMP9 (Figure 6D). As a result, NOV overexpression could promote the EMT process, while NOV silencing could suppress it.

### 3.5. Nov Activates the Expression of Notch1 in ICC

Activating of Notch signaling pathway has been reported to be associated with tumor metastasis [34]. Furthermore, NOV associates with the epidermal growth factor-like repeats of Notch1 by CT (C-terminal cysteine knot) domain during myoblast differentiation [35]. To determine whether NOV expression is involved in Notch signaling pathway activation in ICC cells, we conducted Western blot to detect Notch1 expression, an active protein in the Notch signaling pathway. The results revealed that NOV overexpression enhanced NOTCH1 expression in HCCC-9810 cells (*p* < 0.05), while down-regulated NOV significantly inhibited the expression of NOTCH1 in HCCC-9810 cells (*p* < 0.01) (Figure 7A,B). These results indicated that NOV enhanced the activity of the Notch1 signaling pathway in ICC cells.

### 3.6. miR-92a-3p Targets and Represses NOV Directly

MicroRNAs (miRNA) have long been recognized as a class of important and flexible regulatory molecules [36,37], and their roles in cancer have been extensively studied [38,39]. Herein, to understand the potential role of miRNA in NOV, we queried public databases for NOV-related miRNAs. Five miRNAs were identified: miR-92a-3p, miR-30e-5p, miR-455-5p, miR-197-3p and miR-186-5p. Of these, miR-92a-3p was found in three databases, while miR-30e-5p and miR-455-5p were detected in two databases (Figure 8A). We finally screened these three miRNAs based on integrative results to perform q-PCR validation, but only miR-92a-3p was down-regulated in both HCCC-9810 and RBE cells, suggesting a regulatory relationship with NOV (Figure 8B). Further, the potential regulatory correlations were also verified according to the Starbase database [26] (Figure 8C), and a significant negative correlation between miR-92a-3p and NOV could be found (Figure 8D).

Experimental validation confirmed that miR-92a-3p overexpression had no significant effect on NOV mRNA (Figure 8E), but this miRNA had a significant inhibitory effect on NOV protein (Figure 8F). To verify the direct regulatory relationship between miR-92a-3p and NOV, a dual-luciferase reporter assay was performed. Based on bioinformatics analysis, a potential binding site of miR-92a-3p was located on 343–349 bp of 3′ UTR of NOV (Figure 8G). Compared with the negative control group, the group with cotransfection of miR-92a-3p mimics and NOV 3′ UTR WT had significantly lower luciferase activity. However, no significant difference was observed between the experimental and control groups (Figure 8H).

### 3.7. miR-92a-3p Inhibits the Migration and Invasion via Targeting NOV

Additional experimental validation was performed to discuss whether miR-92a-3p was a crucial regulator of NOV migration and invasion. The migration effect caused by NOV overexpression was rescued by miR-92a-3p overexpression (Figure 9A). Compared with NOV overexpression, cells co-transfected with miR-92a-3p and NOV had significantly weaker invasion ability (Figure 9A–C). Moreover, miR-92a-3p down-regulated E-cadherin and also rescued the upregulation of N-cadherin, vimentin, MMP9, and Snai proteins caused by NOV overexpression (Figure 9D). Similarly, miR-92a-3p also increased expression of Notch1 caused by overexpression of NOV (Figure 9E), indicating that miR-92a-3p could regulate NOV, and then influence cell migration and invasion via targeting NOV.

## 4. Discussion

Several members in the CCN gene family have been implicated in critical biological processes, most notably cancer hallmarks [40,41]. For instance, WISP1, WISP2, and WISP3 are members in the insensitivity to antigrowth signals, self-sufficiency in growth signals, tissue invasion and metastasis, and CTGF is a member in self-sufficiency in growth signals and tissue invasion and metastasis, while NOV is a member in the insensitivity to antigrowth signals. Although homologous members of the CCN gene family have been studied due to their roles in cancers, herein, we find that only five of them are dominantly expressed via a pan-cancer analysis. Dynamic expression patterns indicate that they may be critical in cancer occurrence and development. According to dominant expression patterns, we finally screen the NOV gene as a candidate gene in cholangiocarcinoma to further validate its potential biological role in migration and invasion. Indeed, NOV is detected as having abnormal expression patterns in only CHOL (significant upregulation) and BRCA (significant down-regulation) although more cancers are involved if paired analysis is performed (Figure 1). NOV has a known role in breast cancer and prostate cancer bone metastasis [42,43]. Compared with other homologous genes, NOV is dominantly expressed and second only to CTGF, which is stably expressed in cholangiocarcinoma. Although NOV has been implicated in a variety of biological processes, it is unclear whether it also contributes to the occurrence and development of cholangiocarcinoma. Therefore, this study aims to validate the potential biological role of NOV in cholangiocarcinoma cell lines, particularly its role in cell migration and invasion.

Numerous experimental validations have revealed that NOV contributes to cell migration and invasion. Screening the detailed regulatory mechanism is also important. Additional research has concluded that miRNAs are crucial small regulatory molecules that regulate gene expression, have been implicated in cancer pathology, and may serve as potential prognostic signatures [44,45,46]. Based on miRNA:mRNA interactions, we identify miR-92a-3p as a potential regulatory molecule of NOV. Both bioinformatics analysis and experimental validation have demonstrated that miR-92a-3p can regulate the expression level of NOV protein, affecting subsequent cell migration and invasion processes. miR-92a-3p has been implicated in a variety of biological processes. For instance, the HNF1A-AS1/miR-92a-3p axis affects the radiosensitivity of non-small cell lung cancer by competitively regulating the JNK pathway [47], and cell proliferation is induced in renal cell carcinoma through miR-92a-3p upregulation by targeting FBXW7 [48], miR-21-5p, and miR-92a-3p may be potential biomarkers for hepatocellular carcinoma screening [49], and circulating serum exosomal miR-92a-3p can be as a novel biomarker for early diagnosis of gastric cancer [50]. All of these findings implicate that miR-92a-3p is critical in multiple processes, and our study revealed that miR-92a-3p inhibits NOV expression via a 3′UTR binding site involving cell migration and invasion. 

Furthermore, NOV is also a member of the Notch signaling pathway [35], which may play a potential role in mediating cell adhesion with potential crucial roles in tissue remodeling [51]. In addition, the Notch signaling pathway exhibits a role in cell proliferation, differentiation and fate determination in various tissues, and therefore contributes to cancer pathology [52,53,54]. Herein, to validate whether NOV mediates cell migration and invasion via the Notch signaling pathway, Western blot is used to validate potential interactions. Our results indicate that elevated NOV expression stimulates higher expression of Notch1 protein in HCCC-9810 cells while silencing NOV results in Notch1 protein down-regulation. These results suggest that NOV-Notch1 positively affects Notch signaling and promotes cell migration and invasion. Indeed, CCN3/NOV inhibits BMP-2-induced osteoblast differentiation by interacting with BMP and Notch signaling pathways [55], and NOV overexpression is associated with Notch1 extracellular domain and inhibits myoblast differentiation via the Notch signaling pathway [35]. Moreover, NOV is involved in cartilage protection via PI3K/AKT/mTOR pathway [56]. These findings also confirm the potential role of NOV, particularly as a potential drug target in future cancer treatment.

Overall, based on bioinformatics and experimental validation, our results have revealed that miR-92a-3p-mediated NOV overexpression can promote cell migration and invasion of cholangiocarcinoma, implying its potential role in tumorigenesis. Additional studies should focus on an in-depth analysis of interactions between NOV and other molecules, including interactions with other genes with potential synthetic lethal interactions and interactions with other non-coding RNAs with complex coding-non-coding regulatory networks. 

## Figures and Tables

**Figure 1 genes-12-01659-f001:**
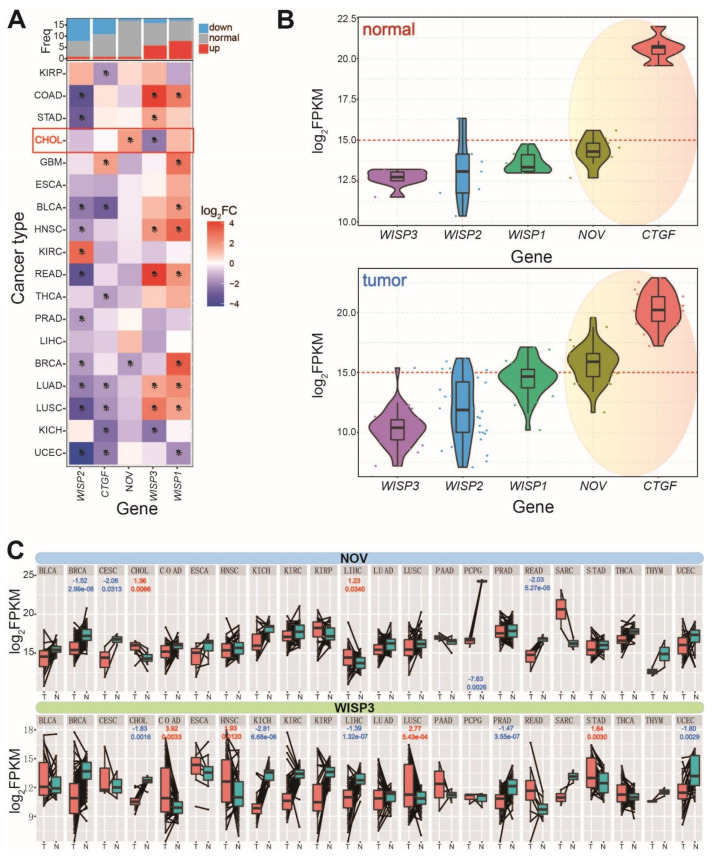
Pan-cancer analysis of expression pattern of CCN gene family. (**A**) Expression distributions of homologous genes in the CCN gene family. * indicates significantly deregulated genes (|log_2_FC| > 1.5, padj < 0.05). (**B**) Expression patterns of five genes in normal and tumor samples in CHOL. (**C**) Paired analysis of deregulated NOV and WISP3 across diverse cancer types. The abnormally expressed genes are presented (|log_2_FC| > 1.2, padj < 0.05 for paired analysis).

**Figure 2 genes-12-01659-f002:**
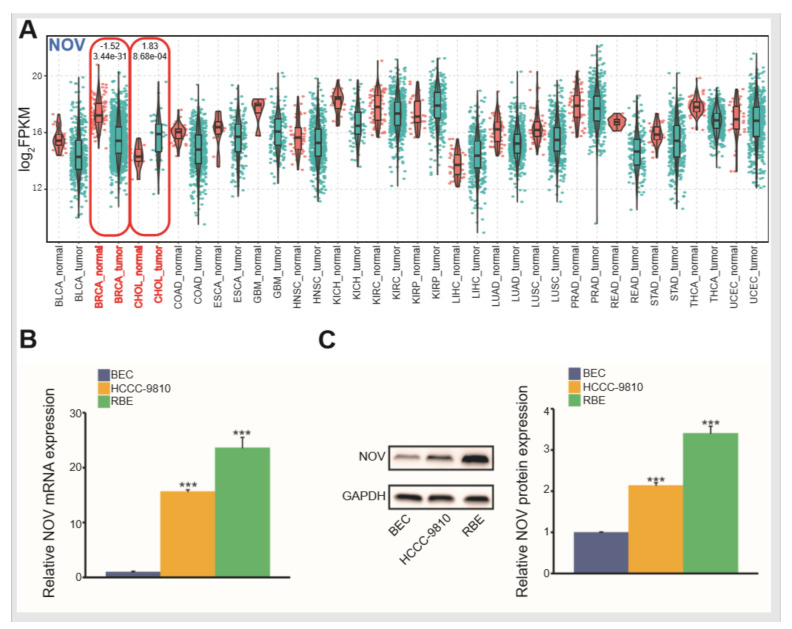
Expression analysis and experimental validation. (**A**) Pan-cancer analysis of screened NOV across cancer types based on tumor and normal samples. The red circled part means that there are significant difference on the expression of NOV in BRCA and CHOL (Tumor vs. Normal) (**B**) NOV mRNA expression based on the q-PCR method. *** indicates *p* < 0.001. (**C**) NOV protein expression based on Western blot. *** indicates *p* < 0.001.

**Figure 3 genes-12-01659-f003:**
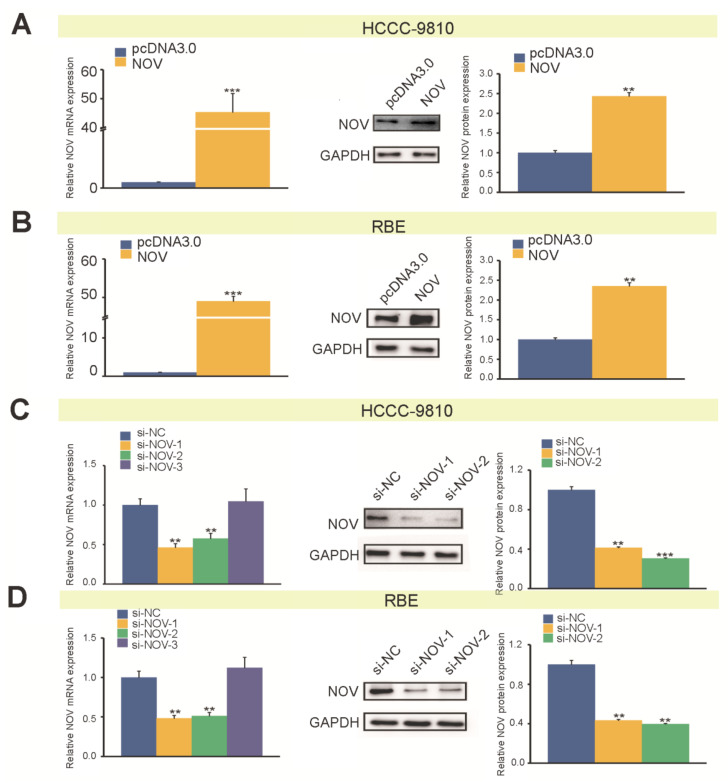
Overexpression and silencing of NOV in HCCC-9810 and RBE. (**A**). Expression efficiency of NOV mRNA (left) and protein (right) in HCCC-9810. (**B**) Expression efficiency of NOV mRNA (left) and protein (right) in RBE. (**C**) Gene silencing of NOV mRNA (left) and protein (right) in HCCC-9810. (**D**). Gene silencing of NOV mRNA (left) and protein (right) in RBE. All experiments are repeated three times independently, and the data are analyzed by *t*-test, ** indicates *p* < 0.01, *** indicates *p* < 0.001.

**Figure 4 genes-12-01659-f004:**
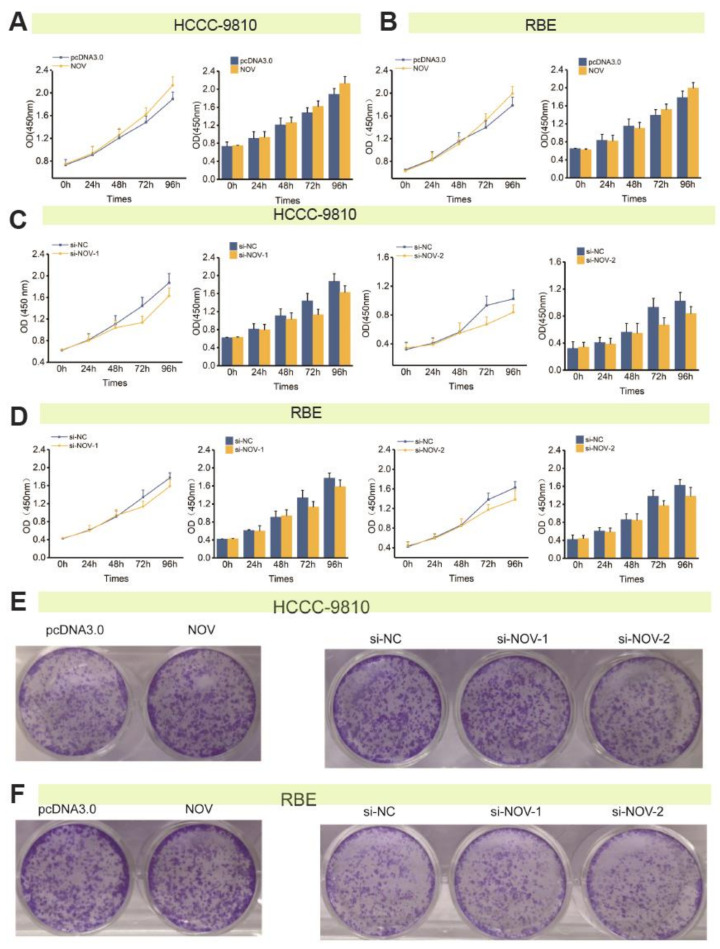
NOV regulates the proliferation of ICC cells. (**A**) The effect of NOV overexpression on cell proliferation via CCK-8 method in HCCC-9810. (**B**). The effect of NOV overexpression on cell proliferation via the CCK-8 method in RBE. (**C**) The effect of NOV silencing on cell proliferation via CCK-8 method in HCCC-9810. (**D**) The effect of NOV silencing on cell proliferation via the CCK-8 method in RBE. (**E**) The effect of NOV overexpression and silencing on cell proliferation via colony-forming assay in HCCC-9810. (**F**) The effect of NOV overexpression and silencing on cell proliferation via colony-forming assay in RBE.

**Figure 5 genes-12-01659-f005:**
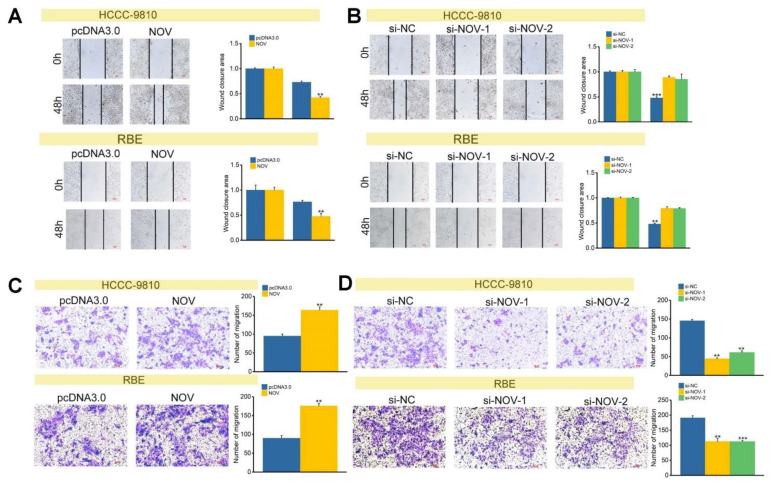
NOV promotes the migration of ICC cells. (**A**) The effect of NOV overexpression on cell migration via wound healing assay. (**B**) The effect of NOV silencing on cell migration via wound-healing assay. (**C**) The effect of NOV overexpression on cell migration via transwell migration (100×). (**D**) The effect of NOV silencing on cell migration via transwell migration (100×). ** indicates *p* < 0.01 and *** indicates *p* < 0.001.

**Figure 6 genes-12-01659-f006:**
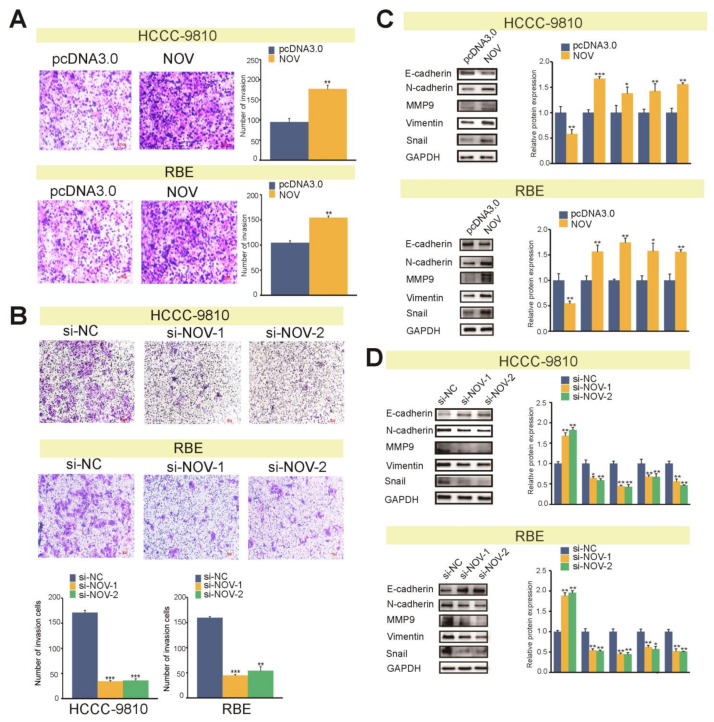
NOV promotes the invasion of ICC cells. (**A**) The effect of NOV overexpression on invasion ability (100×). (**B**) The effect of NOV silencing on invasion ability (100×). (**C**) The effect of NOV overexpression on EMT proteins confirmed by Western blot. (**D**) The effect of NOV silencing on EMT proteins confirmed by Western blot. All experiments are repeated three times independently, and the data are analyzed by *t*-test. * indicates *p* < 0.05, ** indicates *p* < 0.01 and *** indicates *p* < 0.001.

**Figure 7 genes-12-01659-f007:**
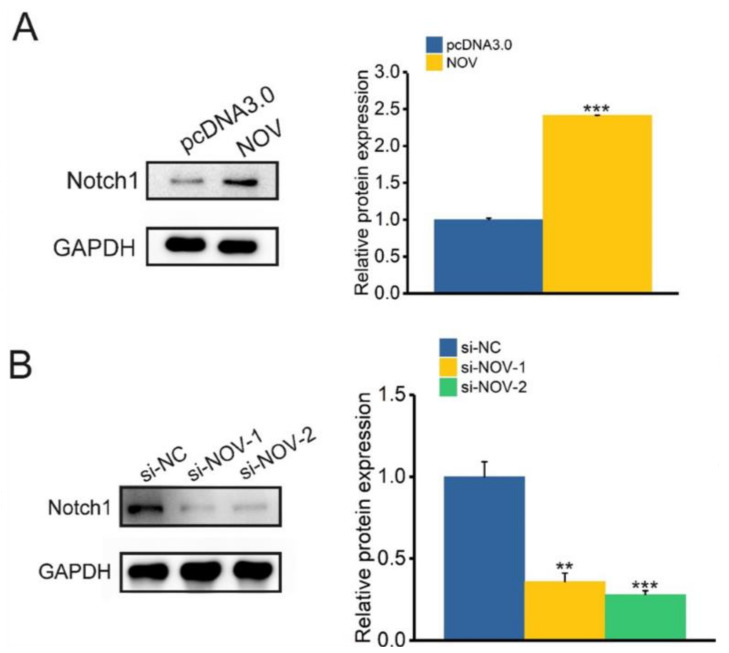
The effect of NOV on the Notch1 pathway. (**A**) The effect of NOV overexpression on Notch1 protein. (**B**) The effect of NOV silencing on Notch1 protein. ** indicates *p* < 0.01 and *** indicates *p* < 0.001.

**Figure 8 genes-12-01659-f008:**
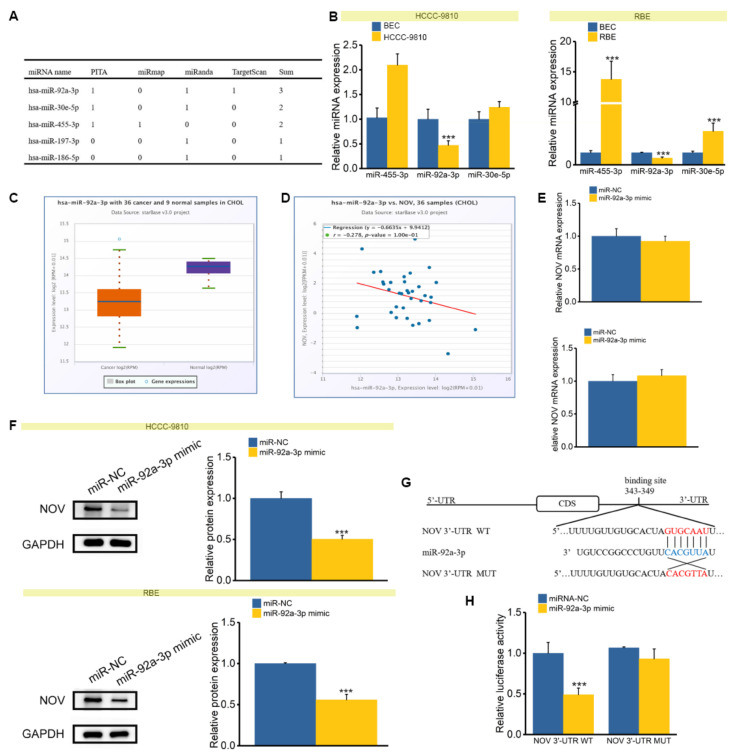
NOV is the direct target of miR-92a-3p. (**A**) Related miRNAs of NOV via different prediction methods. (**B**). Expression levels of miR-92a-3p, miR-30e-5p and miR-455-5p in HCCC-9810 (left) and RBE (right) cells. (**C**) Relative expression level of miR-92a-3p in tumor and adjacent normal samples. (**D**) Expression correlation of miR-92a-3p and NOV. (**E**) The effect of miR-92a-3p on NOV mRNA in HCCC-9810 (left) and RBE (right) cells. (**F**) The effect of miR-92a-3p on NOV protein in HCCC-9810 (up) and RBE (down) cells. (**G**). The binding site of miR-92a-3 and NOV (3′UTR WT and 3′UTR MUT). (**H**) Luciferase activity measurements of WT and MUT after cotransfection of NOV 3′UTR WT or NOV 3′UTR MUT and miR-92a-3p mimic or miR-NC. *** indicates *p* < 0.001.

**Figure 9 genes-12-01659-f009:**
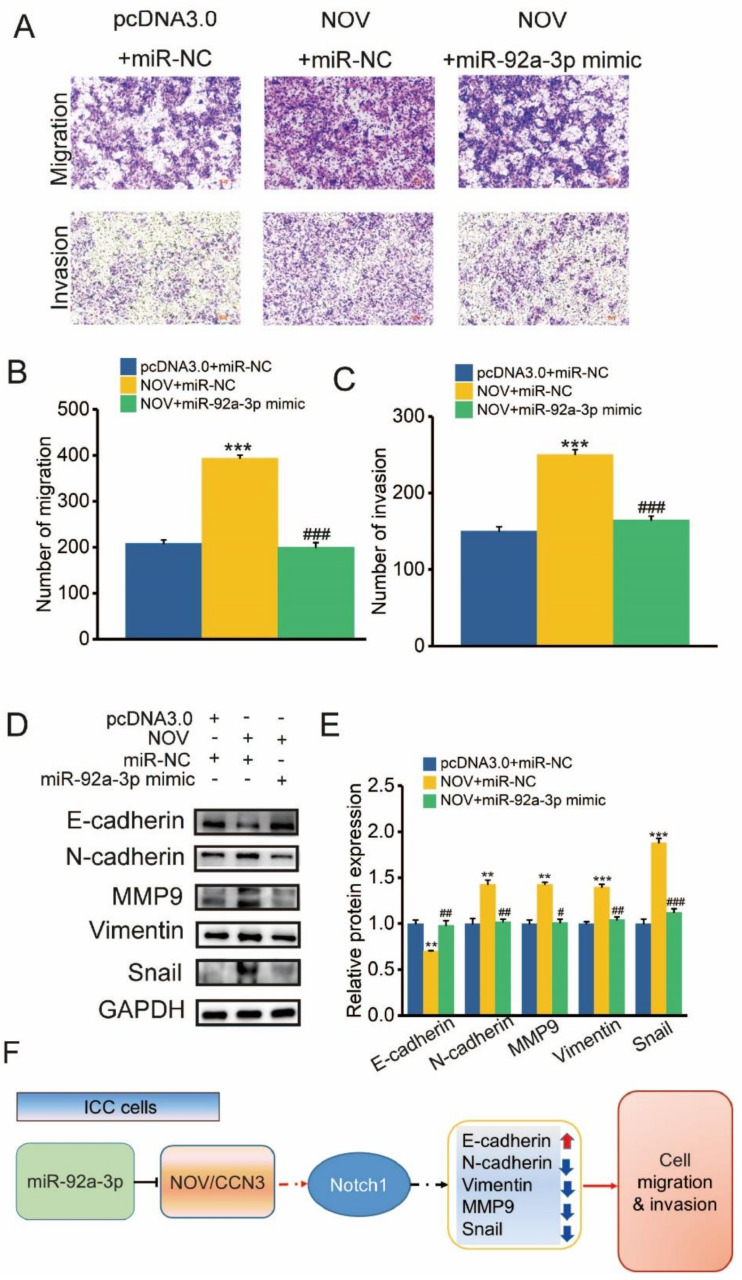
miR-92a-3p contributes to cell transformation and migration mediated by NOV. (**A**). Transwell analysis of migratory and invasive ability after transfection of miR-92a-3p mimics and NOV overexpression plasmid in HCCC-9810 cells. (**B**–**C**) The number of migratory (**B**) and invasive ability (**C**) after transfection of miR-92a-3p mimics and NOV overexpression plasmid in HCCC-9810 cells. (**D**–**E**) Expression levels of E-cadherin, N-cadherin, MMP9, vimentin and Snail after transfecting miR-92a-3p mimics and NOV overexpression plasmid in HCCC-9810 cells based on Western blot. (**F**) Schematic figure indicating the miR-92a-3p /NOV/Notch1 regulatory pathways and resulting biological effects in ICC. pcDNA3.0 + miR-NC vs. NOV + miR-NC: ** indicates *p* < 0.01, *** indicates *p* < 0.001; NOV + miR-NC vs. NOV + miR-92a-3p mimic: ^#^ indicates *p* < 0.05, ^##^ indicates *p* < 0.01, ^###^ indicates *p* < 0.001.

## Data Availability

The data and all materials are available upon request.

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
