# Peer review of "NOV/CCN3 Promotes Cell Migration and Invasion in Intrahepatic Cholangiocarcinoma via miR-92a-3p"

_genes, 2021, doi:10.3390/genes12111659_

Round 1

Reviewer 1 Report

The current manuscript focuses on intrahepatic cholangiocarcinoma (ICC) with the need to investigate potential molecular mechanisms underlying gene diagnosis and therapy. It investigates the role of NOV gene in ICC tumorigenesis suggesting it to be a potential target for diagnosis of ICC.  

In all, the manuscript reads well and improvements could be made based on the following comments:   

1)      Please correct the following spelling mistakes in the manuscript: Line 87: providing,  Line 304:wound healing, Line 346: detect (cancel detected).  

2)      Since the author has used data from reference papers to gather differentially expressed genes, it would be great if the author combines the expression analysis (materials and methods: 2.2) with the data resources in 2.1. 

3)      Materials and methods section 2.6: Please mention what was used to stain the cells after cell fixation (line 136).   

4)      For fig 3, the author has overexpressed NOV in HCCC-9810 and RBE cells. Along with the overexpression of the NOV gene in these cell lines, the author is also testing the effect of silencing of NOV. Although the cell background remains the same, the western blot results for HCCC-9810 for the controls in both the experiments varies (western blot for Fig 3A and Fig 3C). If the empty vector are being used as controls in the same cell line, the expression levels of NOV in the control lanes should match. Please find a better blot for this figure, to avoid confusion.  

5)   On a similar note as point #4: For figure 5, the gap closure after 48hrs (Fig 5A and Fig 5B) a difference is observed in cell migration with just the controls (pcDNA3.0 and si-NC)? Since the cell line remains the same, the addition of controls (empty vectors) to the cells should more or less have the similar effect. Please try to provide different images for this figure. And also please look into Fig 5C, with the same cell background the invasion assay results for the controls should be more or less equal. 

6) Figure 6: For the invasion experiment (fig 6A and fig 6B) the controls here don’t match. Values for number of invasion of HCCC-9810 +pcDNA3.0 and HCCC-9810 + si-NC should be more less same as they are being expressed in same cell background. Please look into this and change the figure as required.  

Reviewer 2 Report

The authors investigated the role of the NOV / CCN3 gene in the regulation of the oncogenic activity of cholangiocarcinoma. The gene was selected based on high-throughput sequencing data available in the Cancer Genome Atlas. The effect of overexpression of the NOV gene and its siRNA knockdown was evaluated in several in vitro assays (cell viability, migration and invasion). The regulatory effect of miR-92a-3p on NOV protein expression and cell biology of HCCC-9810 has also been tested. The authors concluded that NOV / CCN3 may serve as a potential marker and / or therapeutic target for cholangiocarcinoma. The research topic is interesting, the design is appropriate, but the methods are not well described and the results are poorly presented.

There are some comments to each part of manuscript:

Title.

  1. Authors described regulatory pathway: miR-92a-3p > Notch > NOV. However, the title of manuscript is stating that NOV promotes cells migration via miR-92a-3p. So, the direction of the regulation is not clear.

Introduction.

  1. The gene CCN family is wrongly called as CNN several times through the text (pp. 12. 23, 64 …. )
  2. Some abbreviations (HSC, NOV) need to be explained when introduced first time.

Material and methods

  1. (2.3. – 2.4.) It is not clear what kind of media was used (DMEM1640?)
  2. (2.4.) Author should clarify if transfection was stable or transient? If it was transient, it is necessary to indicate when mRNA / protein analysis was performed (immediately, some days after transfection..)
  3. (2.9.) Method of expression data normalization should be presented (dCt, ddCt)
  4. (2.10). Fragment of text pp 192 -200 is not relevant to this manuscript

Results

  1. (3.1.) Images in Figure 1C require explanation. What is meaning of these nice figures? What kind of cancer types were included in analysis? If expression difference between normal and cancer tissues is shown by pink and green bars, what is meaning of black figures behind of bars?

2 (3.2.) Method of expression data normalization should be described in legend of Figure 2B.

3 (3.3.) Figure 3.: It is not clear what kind of statistic is shown in Figure? Results of several transfections? Results of several RT-PCR / Western blots? What was a method of statistical significance evaluation?

  1. Figure 4A-B-C-D and Figure 5A-B are hardly visible. It is probably better to present one figure enlarged and other data can be presented graphically.
  2. Figure 6B. What does it mean “Number of invasion”?
  3. Figure 6C-D It is not clear what kind of statistic is shown in Figure? Results of several Western blots? What was a method of statistical significance evaluation?
  4. Section 3.6. and 3.7. described role of miR-92a in regulation of NOV and tumorigenic potency of HCCC-9810 cells. It will be much easily for reader to see some kind of schematic figure indicating the regulatory pathways and resulting biological effects.

Discussion.

Authors focused their research on cholangiocarcinoma because of results of in silico data analysis. However, differential expression of NOV is observed in many other types of cancer (as it is shown in Figure 1C). More general role of this regulatory pathway (not only ICC-specific) should be discussed.

Moreover, function of NOV protein and its possible role in cell biology should be discussed.

Round 2

Reviewer 1 Report

Thanks for addressing the comments.

Reviewer 2 Report

Authors revised manuscript in according to suggestion. It can be accepted now for publication.